# Determination of the Characteristics of Non-Stationary Random Processes by Non-Parametric Methods of Decision Theory

**Bulat-Batyr Yesmagambetov** 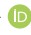

Department of Automation, Telecommunications and Management, M. Auezov South Kazakhstan University, Shymkent 160012, Kazakhstan; bulatbatyr@mail.ru

**Abstract:** This article is devoted to methods of processing random processes. This task becomes particularly relevant in cases where the random process is broadband and non-stationary; then, the measurement of a random process can be associated with an assessment of its probabilistic characteristics. Very often, a non-stationary broadband random process is represented by a single implementation with a priori uncertainty about the type of distribution function. Such random processes occur in information and measuring communication systems in which information is transmitted at a real-time pace (for example, radio telemetry systems in spacecraft). The use of methods of traditional mathematical statistics, for example, maximum likelihood methods, to determine probability characteristics in this case is not possible. In addition, the on-board computing systems of spacecraft operate under conditions of restrictions on mass-dimensional characteristics and energy consumption. Therefore, there is a need to apply accelerated methods of processing measured random processes. This article discusses a method of processing non-stationary broadband random processes based on the use of non-parametric methods of decision theory. An algorithm for dividing the observation interval into stationary intervals using non-parametric Kendall's statistics is considered, as are methods for estimating probabilistic characteristics on the stationary interval using ordinal statistics. This article presents the results of statistical modeling using the Mathcad program.

**Keywords:** random process; non-parametric statistics; Kendall's statistics; ordinal statistics; stationary interval; probability characteristics

## 1. Introduction

This article is devoted to the processing of non-stationary broadband signals in radio communication systems, such as the radio telemetry systems of spacecraft (RTSSs). RTSSs have features that distinguish them from other radio communication systems [1–4]. The first feature of RTSSs is that the signals measured by the on-board system must be transmitted to Earth at a real-time rate. This feature is due to the fact that the measured parameters must be monitored on Earth in real time in order to be able to make operational management decisions in case of any emergency situations. Examples of such parameters are the temperature and pressure in the combustion chamber of a space rocket. These parameters are converted into electrical signals and transmitted via communication channels to Earth in real time. If the change in these parameters in time does not occur as it should, appropriate decisions are made on Earth. For example, the engines can be disconnected from the fuel supply so that the space complex does not catch fire. The second feature of RTSSs is that the measured parameters (in our case, temperature and pressure in the combustion chamber) are, as a rule, a non-stationary broadband (rapidly changing) random process (NSBRP). Such a process is always represented by a single implementation in conditions of a priori uncertainty about the form of the distribution function (this is especially true if the change in the measured parameter does not occur normally, i.e., in emergency mode). The third feature of RTSSs is the restriction on the mass-dimensional characteristics and power consumption of the on-board computing systems (CSs). These features of RTSSs require

special algorithms for processing the measured parameters. For example, it is impossible to process rapidly changing (broadband) signals using the traditional cyclic sampling method followed by the use of classical data compression methods [5,6]. This is due to the fact that, due to the high frequency of signal changes, there is a large load on the communication channel, the bandwidth of which is always limited [7,8]. In addition, a large load will be exerted in this case on the on-board CS, the speed and memory capacity of which are always limited due to restrictions on mass-dimensional characteristics and power consumption. If we take into account that the number of measured sources of information can be several tens or even hundreds [9], then it becomes clear that the load on both the communication channel and the CS of an RTSS will be very large.

Therefore, when processing fast-changing (broadband) signals in RTSSs, as a rule, they are limited to calculating probabilistic characteristics and transmitting them via communication channels to Earth. This approach to the processing of non-stationary broadband signals allows, on the one hand, to bring the volume of transmitted data in line with the bandwidth of the communication channel and, on the other hand, to reduce the requirements for the performance of on-board CSs. At the same time, the amount of information received will be sufficient to make a correct judgment about the measured process. Methods of classical mathematical statistics are used to calculate the probabilistic characteristics of random processes (RPs). But, in this case, they turn out to be completely unsuitable for the following reasons: Firstly, to calculate the mean and variance, a priori knowledge of the type of distribution function of the measured RP is necessary. For example, well-known maximum likelihood methods are effective only for a Gaussian RP and are not effective for other distributions. They turn out to be completely ineffective if the type of distribution function is not known a priori. Secondly, to apply the methods of traditional mathematical statistics, an ensemble of implementations is required (in theory, at least 1000 implementations; in practice, more than 100). Thirdly, the apparatus of mathematical statistics is designed for stationary random processes (SRPs) and cannot be applied if the random process is non-stationary. In our case, we are dealing with a single implementation of an NSBRP.

Thus, in order to process NSBRPs represented by a single implementation under conditions of a priori uncertainty about the form of the distribution function, it is necessary to use other methods. At the same time, the measured RP should be described by an additive–multiplicative model of the following type:

$$y(t) = F(t) + X(t), \tag{1}$$

where $F(t)$ is the non-stationary average of the measured random process, and $X(t)$ is the SRP (Figure 1).

In the figure, $y$ (blue ) is a measured non-stationary random process represented by a single implementation; $F(t)$ is the non-stationary average of the measured process; and $x(t)$ (black) is a stationary random process.

To obtain an estimate of $F(t)$, various methods of optimal filtering (for example, a Kalman–Bewsey filter) can be used. However, to build a filtering algorithm, a priori knowledge of the distribution function type and spectral density of the process is necessary. In addition, filtration methods do not allow obtaining estimates of other probabilistic characteristics of the stationary component. Such a setting of the task may be sufficient in cases where only information about the average value of $F(t)$ is needed, but for the purposes of complete processing, it is necessary to obtain information about the random component of the $X(t)$ process.

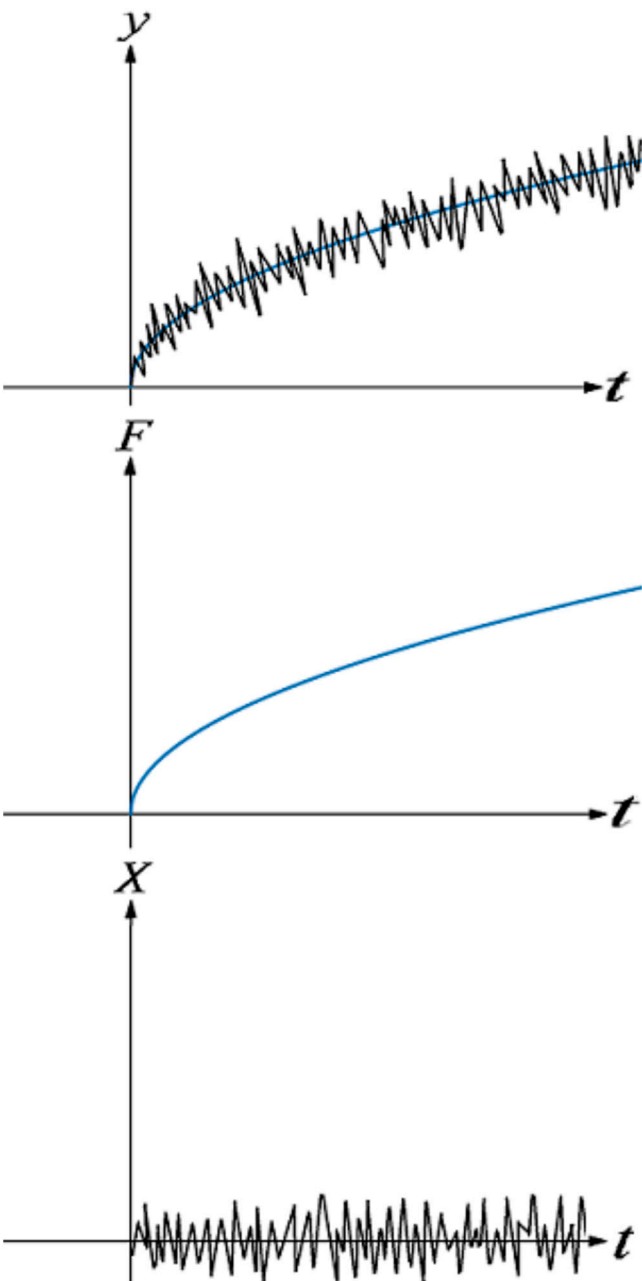

**Figure 1.** Non-stationary random process model.

Various methods of mathematical statistics based on regression analysis and time series analysis [10] could be used to analyze RPs. But to use these methods, it is required that the entire time series be known in advance. Only in this case can the characteristic equations of the autoregressive time series model be solved and their roots found. We are considering a non-stationary random process (NSRP) that is formed at the rate of data receipt. In other words, at each moment of time, only those samples of the measured process that are received at this moment in time and at previous moments of time are available for analysis. The entire time series becomes known only at the end of the processing of an RP. This is the specificity of the RP under consideration. For this reason, the methods of processing RPs used in various fields, such as, for example, finance and climatology [11], cannot be used.

In such cases, it is possible to construct algorithms for estimating the probabilistic characteristics of an NSRP using non-parametric statistics [12–14].

It is known [15,16] that the non-parametric statistics (NPS) call some function of a random variable with an unknown probability distribution. This function itself has a known distribution, the properties of which in some way characterize the properties of an unknown distribution of the original random variable. Knowing the distribution of NPS, we can use it to formulate and test different hypotheses about the properties of unknown distributions (for example, their symmetry, stationarity, and so on).

## 2. Non-Parametric Statistics and Their Use for Processing Random Processes

Consider the most common NPS.

Let $Y = \{y_1, y_2, \ldots y_n\}$ be a vector of sample values from the process $y(t)$, obtained by sampling it in time in an interval of $\Delta t$ with $\Delta t > \tau_k$, where $\tau_k$ is the correlation interval of the process. Let us determine the sign function of observations in the form

$$sign\ y = \begin{cases} 1, & y \geq 0 \\ -1, & y < 0 \end{cases} \tag{2}$$

We introduce a unit jump function or a positive sign vector

$$u(y) = \begin{cases} 1, & y \geq 0 \\ 0, & y < 0 \end{cases} \tag{3}$$

related to the sign function by the relation

$$2u(y) = sign(y) + 1.$$

Functions (2) and (3) are called sign statistics or elementary inversions, and the vector

$$\overline{U}(y) = \{u_1(y),\ u_2(y), \ldots, u_n(y)\},$$

composed of sign statistics, is called a sign vector.

The distribution of sign statistics is binomial, with parameter $n$ equal to the sample size:

$$P_n(u = i) = C_n^i p^i q^{n-i}. \tag{4}$$

The mean and variance of sign statistics are defined as $M_u = np$ and $D_u = npq$, respectively. The parameter $p$ of this distribution is the probability of sign statistics appearing in a single test.

If we rearrange the $\overline{Y}$ sample items in ascending order:

$$\overline{Y} = \{y^{(1)}, y^{(2)}, \ldots \ldots y^{(n)}\}, \tag{5}$$

where $y^{(k)} \leq y^{(j)}$ for $k < j$, then we obtain a vector called the vector of ordinal statistics, and its elements $y^{(k)}$ are ordinal statistics. When replacing the elements of the sample $y^{(k)}$ with their ranks $R_k$, where $R_k = K$ is the ordinal number of the element $y^{(k)}$ in the ranked series, we obtain the vector $\overline{R}(\overline{y}) = \{R_1, R_2, \ldots, R_n\}$, called the rank vector. If we need to have both information about the rank $R$ of the sample value and its ordinal number $i$ in the original sample, then we can enter the designation $R_i$, which means that $R$ is the rank of the $i$-th observation in the sample. It is believed that $n$ is known and fixed.

Consider the nature of specific problems solved using non-parametric methods. First of all, this task of estimating unknown distributions, which differs from the problem of approximating an unknown distribution by known functions, is considered in ordinary statistics. In a non-parametric formulation, this problem can be formulated as an estimate of the difference between an unknown distribution and a given class of distributions. If it is necessary to specify these differences, the task of estimating the parameters of distributions is formulated. In this case, not the parameter itself is evaluated, but the parameter of the difference between distributions within a given non-parametric class. Another category

of non-parametric problems is testing non-parametric hypotheses. In any non-parametric hypothesis testing problem consisting of two competing hypotheses, the alternative is always non-parametric, and the null hypothesis can be either simple or non-parametric. The difference between hypotheses is not related to a specific type of distribution function, since one of the hypotheses has a class of unknown distributions. The essence of the procedure is that, based on the original sample, it is necessary to attach an algorithm, the result of which will be a decision on the truth of one of the hypotheses.

Consider, for example, the procedure for generating decision rules to test the symmetry hypothesis of the distribution of some random variable $y$ (Figure 2), using sign statistics (Function (3)) for this.

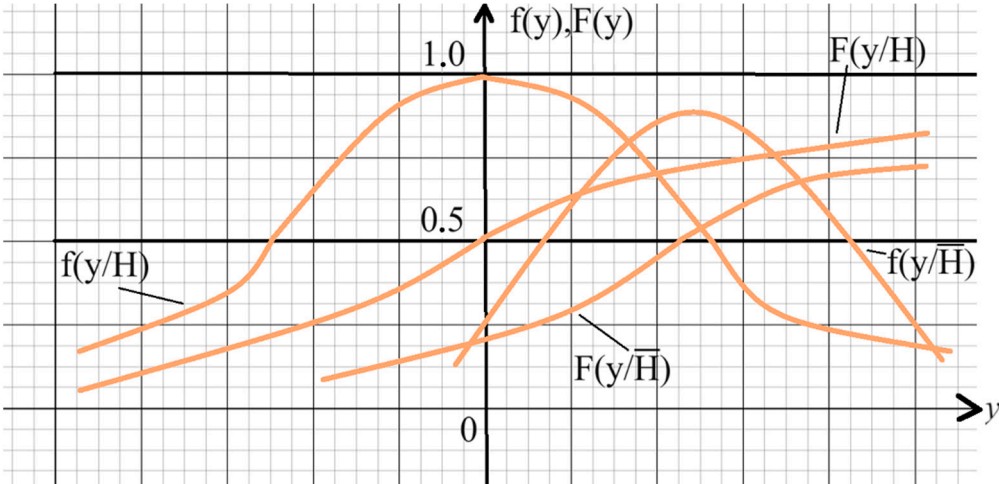

**Figure 2.** Testing the distribution symmetry hypothesis.

In the figure, $f(y/H)$ and $F(y/H)$ are the probability density function and the distribution function of a symmetric distribution; $f(y/\overline{H})$ and $F(y/\overline{H})$ are the probability density function and the distribution function of a non-symmetric distribution.

Take the character counter function into consideration:

$$z = \sum_{i=1}^{n} u_i(y_i)$$

which has a binomial distribution according to Formula (3).

As can be seen from Figure 2, for the symmetrical distribution, $P = 0.5$; and for the asymmetrical distribution, $P \neq 0.5$. In other words, the curve of the distribution function $F(y/H)$ intersects the ordinate axis at point 0.5, and the curve of the distribution function $F(y/\overline{H})$ intersects the ordinate axis at point unequal to 0.5 (lower than 0.5).

We introduce the null hypothesis about the symmetry of the distribution:

$$H : f(y) = f(-y), \ P = 0.5$$

and an alternative hypothesis about asymmetry of the distribution:

$$\overline{H} : f(y) \neq f(-y), \ P \neq 0.5.$$

Given that with sample volumes of $n > 20$, the binomial distribution is well approximated by the Gaussian distribution, the decisive rule on the Neyman–Pearson criterion can be written as follows: if $Z > C_1$, the data are true to the alternative hypothesis; if $Z < C_1$, the data are true to the null hypothesis. At the same time, $C_1$ is the threshold of the decisive rule:

$$C_1 = \frac{Z_\alpha}{2}\sqrt{n} + \frac{n}{2} - 1. \tag{6}$$

The value of the threshold of the decisive rule is selected from the following condition, which can be found in [17,18]:

$$\alpha = 1 - F\left[\frac{C_1 + 1 - \frac{n}{2}}{\frac{\sqrt{n}}{2}}\right]. \tag{7}$$

Here, $\alpha$ is a Gaussian distribution parameter called the significance level (in the literature, this parameter is often called the probability of error of the first kind, or the "probability of false alarm"), and $n$, as already noted, is the sample size.

Such a decisive rule is unbiased only for $P > 0.5$. At $P < 0.5$, the decisive rule $Z < C_2$ turns out to be unbiased, where the threshold

$$C_2 = \frac{Z_{1-\alpha}}{2}\sqrt{n} + \frac{n}{2} - 1. \tag{8}$$

In this case, the probabilities of error of the second kind (signal skipping) are determined from the following relationships:

$$\beta_1 = F\left[\frac{Z_{\alpha/2} - \sqrt{n}(p - 0.5)}{\sqrt{pq}}\right]; \tag{9}$$

$$\beta_2 = F\left[\frac{Z_{1-\alpha/2} - \sqrt{n}(p - 0.5)}{\sqrt{pq}}\right], \tag{10}$$

which are described in [19].

With small volumes of observations ($n < 20$), the value of the $\alpha$ significance level can be determined according to the Bernoulli distribution

$$\alpha = P(y > C_1|H) = \sum_{m=C_1}^{n} C_n^m 0.5^m 0.5^{n-m}, \tag{11}$$

where the value of the $C_1$ threshold is determined. The error amount of the second kind in this case will be determined from the relation

$$\beta_1 = P(y > C_1|\overline{H}) = \sum_{m=0}^{m=C_1} C_n^m p^m (1 - p)^{n-m}, \tag{12}$$

if the distribution parameter $P > 0.5$. In the same case, when $P < 0.5$, the amount of the error of the second kind should be defined as

$$\beta_2 = P(y > C_2|\overline{H}) = \sum_{m=C_2}^{n} C_n^m p^m (1 - p)^{n-m}. \tag{13}$$

Percentage points as well as distributions of various modifications of variable (11) can be found in [20].

## 3. Division of the Observation Interval into Stationarity Intervals Using Kendall's Statistics

We consider the possibility of using non-parametric methods of decision theory to estimate the probabilistic characteristics of the NSRP described by model (1) (Figure 1). By probability characteristics, we mean the mean value, variance (standard deviation), distribution function, and correlation function. Recall that a random process represented by a single implementation is considered in conditions of a priori uncertainty about the type of distribution function. To estimate the probabilistic characteristics of such an RP, it is advisable to first identify a non-stationary average $F(t)$ (obtain an estimate of the average value) and then obtain estimates of other probabilistic characteristics of component $X(t)$.

To increase the accuracy of the separation of the non-stationary component of the random process, it is desirable to divide the entire observation interval into stationary intervals, the length and number of which are determined by the type of non-stationary component $F(t)$ and the probabilistic characteristics of stationary component $X(t)$. To divide the observation interval into stationary intervals, we will use Kendall's statistics, which are well known in the literature [15]:

$$T^2 = \sum_{i=1}^{n-1} \sum_{k=i+1}^{n} u(y_i, y_k) \tag{14}$$

where

$$u(y_i, y_k) = \begin{cases} 1, & y_i \geq y_k \\ 0, & y_i < y_k \end{cases}$$

and are called sign statistics or elementary inversions. Here, $y_i$ and $y_k$ are the values of the measured process obtained by sampling with a sampling interval of $\Delta t$. The sampling interval in this case is selected based on the statistical independence of the two adjacent sample values $y_i$ and $y_{i+1}$, that is, $\Delta t \geq \tau_k$, where $\tau_k$ is the correlation interval of the random process. The selection of the sampling interval is a separate task that needs to be solved.

Using Kendall's statistics makes it quite easy to divide the time series of observations $y(t)$ by the finite number of stationary intervals with a given probability $P = 1 - \alpha$ by parameters such as the average value $m[y(t)]$ and variance $D[y(t)]$. Here, $\alpha$ is the probability that the interval is not stationary. In the Russian literature, $\alpha$ is commonly referred to as "the probability of a false alarm" or "the level of significance". The division procedure consists of calculating the current values $T^2$ and the permissible limits $T_{min}^2[i; 1 - \alpha/2]$ and $T_{max}^2[i; \alpha/2]$ and checking the stationarity condition by the Neyman–Pearson criterion [21–23]:

$$T_{min}^2 < T_i^2 \leq T_{max}^2. \tag{15}$$

The distribution of the Kendall variable for sample sizes $n > 10$ differs little from the Gaussian distribution [19].

The values of the permissible limits of the decision rule thresholds can be determined from the relations

$$T_{min}^2 = M\left[T^2\right] - x_{\alpha/2}\sqrt{D[T^2]}, \tag{16}$$

$$T_{max}^2 = M\left[T^2\right] + x_{\alpha/2}\sqrt{D[T^2]}, \tag{17}$$

where $x_{\alpha/2}$ is the percentage point of the Gaussian distribution.

To calculate $T_{min}^2$ and $T_{max}^2$, it is necessary to know the average value and variance of Kendall's statistics. The formulas for calculating the mean and variance of Kendall's statistics are known to the author, but they are know-how and therefore not given in this article. The value of the percentage point is taken for the Gaussian distribution, since the binomial distribution is well approximated by the Gaussian distribution.

Kendall's statistics are symmetrical around their mathematical expectation since they are indifferent to how elementary inversions are obtained, whether it is by fulfilling the inequality $y_i < y_j$ or $y_j < y_i$ for $j = i + 1, i + 2, i + 3 \ldots$. This fact means that, in many practical applications, a reverse procedure can be used, which gives tangible advantages in efficiency and other indicators.

The reversibility of the procedure can be used to divide time series into stationary intervals, at which the line of current values of $T^2$ is sequentially reflected from permissible boundaries. In this case, it is necessary to constantly take into account the moments of transition of the sign function $u(y)$ to the opposite value, that is, to fix the reflection points of total inversions from permissible boundaries. As in previous cases, non-stationary measurement data are divided into stationary at some intervals, the statistical characteristics

of which are constant but not equal to each other. Consider the method of reflected inversions in more detail (Figure 3).

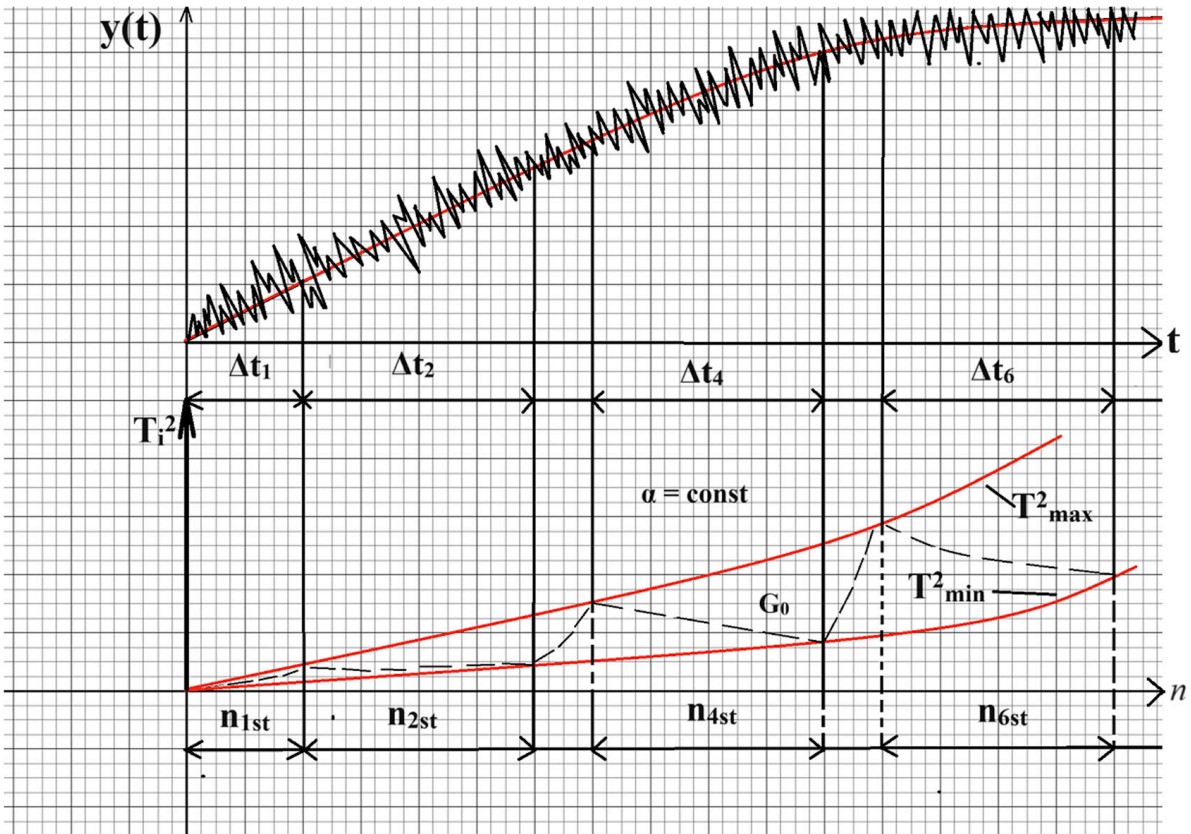

**Figure 3.** Division of observation interval into stationarity intervals.

According to incoming samples $y_1, y_2, \ldots y_n$ of the measured series, $y(t)$ calculates the function $u(y_i, y_j)$ from which Kendall's statistics are determined. Valid bounds $T^2_{max}$ and $T^2_{min}$ are defined for a given significance level $\alpha$. In addition to the methods described above, a comparison between $T^2_i$ and both $T^2_{max}$ and $T^2_{min}$ is conducted, as a result of which there can be two outcomes:

1. Inequality (15) is performed, and the process does not leave the field of stationarity;
2. Inequality (15) is broken, and the process leaves the field of stationarity.

The point corresponding to the moment of crossing line $T^2_i$ from one of the permissible boundaries is fixed, and the sign function $u(y_i, y_j)$ is "flipped" to the opposite value, as a result of which a fracture point is formed on line $T^2_i$ and the calculation process is repeated. When the second of the permissible boundaries is reached, the function $u(y_i, y_j)$ is flipped again while fixing the fracture point on line $T^2_i$. Thus, line $T^2_i$ is inside the stationary area all the time and is consistently reflected from the permissible boundary lines.

After determining the stationarity intervals, the probabilistic characteristics of the measured NSRP are evaluated. Evaluations are carried out at each stationarity interval separately.

In the figure, $\Delta t_i$ are the intervals of stationarity; $n_{ist}$ is the number of samples at each interval of stationarity; $G_0$ is the region of stationarity; and $T^2_{min}$ and $T^2_{max}$ are the boundary values of the region of stationarity black dotted lines—variable $T^2_i$.

### 4. Evaluation of Probabilistic Characteristics of a Random Process Using Ordinal and Rank Statistics

The simplification of estimates of probabilistic characteristics is possible when using ordinal statistics (OS) of a ranked series when ranking the data obtained on the stationarity interval in decreasing or increasing order:

$$x_{(1)} \leq x_{(2)} \leq \cdots \leq x_{(R)} \leq \cdots \leq x_{(N)} \tag{18}$$

In a number of works [24,25], studies of errors in estimating probabilistic characteristics by ordinal statistics were carried out. However, these works were limited to the study of a stationary stochastic process, while obtaining estimates of probabilistic characteristics from samples of a non-stationary stochastic process is of particular interest.

The application of ordinal statisticians allows for the use of simple enough procedures for the average estimation based on central ordinal statistics (COS), ranked beside [26,27].

$$\widetilde{m}_{11} = x_{(c)}; \ \widetilde{m}_{12} = x_{(c+1)}; \ \widetilde{m}_{21} = \frac{1}{2}\left(x_{(c-1)} + x_{(c)}\right);$$

$$\widetilde{m}_{22} = \frac{1}{2}\left(x_{(c)} + x_{(c+1)}\right); \ \widetilde{m}_{2j} = \frac{1}{2}\left(x_{(c)} + x_{(c+j)}\right);$$

$$\widetilde{m}_{31} = \frac{1}{3}\left(x_{(c-1)} + x_{(c)} + x_{(c+1)}\right).$$

There are estimates based on the truncation ranked series

$$\widetilde{m}_{41} = \frac{1}{N-2}\sum_{i=2}^{N-2} x_{(i)},$$

$$\widetilde{m}_{4j} = \frac{1}{N-j}\sum_{i=j}^{N-j} x_{(i)};$$

and also using extreme ordinal statistics:

$$\widetilde{m}_{51} = \frac{1}{2}\left(x_{(N)} + x_{(1)}\right); \ \widetilde{m}_{52} = \frac{1}{2}\left(x_{(N-1)} + x_{(2)}\right);$$

$$\widetilde{m}_{5j} = \frac{1}{2}\left(x_{(N-j+1)} + x_{(j)}\right).$$

The estimations using various combinations of enumerated estimations can be synthesized as follows:

$$\widetilde{m}_{61} = \frac{1}{2}\left(x_{(K1)} + x_{(K2)}\right),$$

$$K1 = E[0.73 \, N], \ K2 = E[0.27 \, N];$$

$$\widetilde{m}_{62} = \frac{1}{2}\left(x_{(K1)} + x_{(K2)}\right),$$

$$K1 = E[0.75 \, N], \ K2 = E[0.25 \, N];$$

$$\widetilde{m}_{71} = \nu_1 x_{(K1)}; \ \widetilde{m}_{72} = \nu_1 x_{(K1)} + \nu_2 x_{(K2)};$$

$$\widetilde{m}_{7j} = \nu_1 x_{(K1)} + \nu_2 x_{(K2)} + \cdots + \nu_j x_{(Kj)}, \ j \ll N;$$

$$\tilde{m}_{81} = v_{12}(x_{(K1)} + x_{(K2)});$$

$$\tilde{m}_{82} = v_{12}\left(x_{(K1)} + x_{(K2)}\right) + v_{34}\left(x_{(K3)} + x_{(K4)}\right);$$

$$\tilde{m}_{8j} = v_{12}\left(x_{(K1)} + x_{(K2)}\right) + \cdots + v_{ij}(x_{(Ki)} + x_{(Kj)});$$

The most optimal procedure for estimating the mean is an estimate based on the COS of the ranked series [28,29]:

$$\tilde{m}_{11} = x_{(c)}; \tag{19}$$

Obviously, central ordinal statistics are the easiest to implement. In Figure 4, the comparative analysis of computing costs (memory size $S$ and average calculation time $T$) of various modes of estimation of the mean is shown (where $S(\tilde{m}_0)$ and $T(\tilde{m}_0)$ are the memory size and the average calculation time when using the maximum likelihood estimate). The minimum costs, apparently, have estimations of an aspect $\tilde{m}_{11}$.

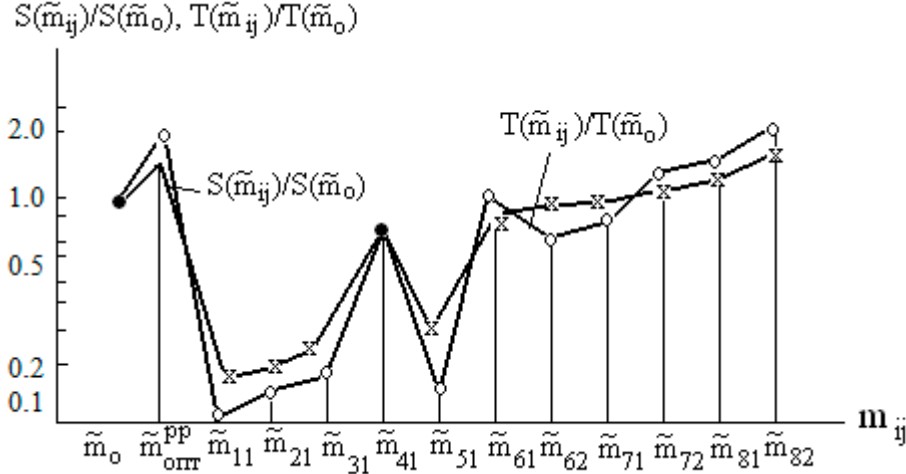

**Figure 4.** Comparative analysis of computing costs of an estimation. $S$—memory size; $T$—average calculation time; $S(\tilde{m}_0)$ and $T(\tilde{m}_0)$—the memory size and the average calculation time when using the maximum likelihood estimate.

When measuring variance, it is advisable to use the same ranked series of the ordinal statistics as when estimating the mean. At the same time, it is best to estimate not the variance of the process itself but the standard deviation. To estimate the standard deviation in non-parametric statistics, the simplest range functions span $W_1 = x_{(N)} - x_{(1)}$ and under the scope $W_j = x_{(n-j+1)} - x_{(j)}$ are used, using the extreme of the order statistics ranked series:

$$\tilde{\sigma}_{11} = v\left(x_{(N)} - x_{(1)}\right),$$

$$\tilde{\sigma}_{12} = v\left(x_{(N-1)} - x_{(2)}\right).$$

It is also possible to use the estimations:

$$\tilde{\sigma}_{3j} = v\left(x_{(K1)} - x_{(K2)}\right),$$

where, for $\sigma_{31}$, $K1 = E[0.75\,N]$, $K2 = E[0.25\,N]$; and, for $\sigma_{32}$, $K1 = E[0.73\,N]$, $K2 = E[0.27\,N]$.

As in the case of estimating the mean, different combinations of the central order statistics and extreme order statistics (EOS) are possible:

$$\tilde{\sigma}_{4j} = \nu\left(x_{(c+j)} - x_{(c-j)}\right), \; j = 1 \div 4.$$

The coefficient $\nu$ can be assigned from a wide range; however, the most effective factor values are as follows:

$$\nu = 1; \ldots 1/2; \ldots 1/3; \ldots 1/4;$$

The optimal estimate of the variance is an estimate of the type

$$\tilde{\sigma}_{11} = \nu\left(X_{(N)} - X_{(1)}\right). \tag{20}$$

The optimal estimation conditions can be written in the following form [27,28]:

$$\tilde{\sigma}_{opt} = \tilde{\sigma}_{11} = \nu\left(X_{(N)} - X_{(1)}\right), \; \begin{cases} \nu = \frac{1}{3}, N < 15 \\ \nu = \frac{1}{4}, N \geq 15 \end{cases} \tag{21}$$

The ranked series of the ordinal statistics can be used to estimate the distribution function $F(x)$ and the probability density function $f(x)$. In this case, it is enough to estimate one of them and indirectly obtain an estimate of the other by differentiating the distribution function $F(x)$ or integrating the probability density function $f(x)$. As for the method of transmission of the telemetry data, it is better to evaluate the distribution function $F(x)$ due to the greater complexity of the implementation of methods for estimating $f(x)$ and the better noise immunity of transmission $F(x)$. Therefore, the consideration of methods of distribution function estimation should be paid more attention.

The classic definition of the distribution function as the probability of the event $(x(t) < x)$ allows us to write the following relation:

$$\tilde{F}_0(x) = Prob(x(t) < x) = \frac{N_x}{N} = \frac{1}{N}\sum c(x - x_i),$$

where $Prob(\ldots)$ means probability; $N$ is the sample size; $N_x$ is the number of samples of the process $x(t)$ not exceeding the value of $x$; and $c(x - x_i)$ is the comparison function.

$$c(x - x_i) = \begin{cases} 1, \; x \geq x_i \\ 0, \; x < x_i \end{cases}$$

The statistical relationship between the sample value and its rank allows us to write the following approximate value:

$$\tilde{F}_1(x) = \tilde{F}_1\left(x_{(R)}\right) = \frac{R}{N+1}$$

A modification of this method, based on fixation as a quantile rather than an order statistic $x(R)$ of rank $R$, involves a linear combination of $Q$-order statistics $x(R)$ of rank $R$:

$$x_{(R)}^Q = \sum_{q=1}^{Q} A_q x_{(q)}$$

which allows us to generate the following estimates:

$$\tilde{F}_2\left(\frac{1}{2}\left(x_{(R-1)} + x_{(R)}\right)\right) = \frac{R}{N+1};$$

$$\tilde{F}_3\left(\frac{1}{2}\left(x_{(R)} + x_{(R+1)}\right)\right) = \frac{R}{N+1};$$

$$\tilde{F}_4\left(\frac{1}{3}\left(x_{(R-1)} + x_{(R)} + x_{(R+1)}\right)\right) = \frac{R}{N+1}.$$

At these estimations, in the capacity of a quantile magnitude, the average of two or three ordinal statisticians is fixed.

Another method for estimating a cumulative distribution function is based on the evaluation of a non-parametric tolerant interval ($L_2$-$L_1$), where $L_1$ and $L_2$ represent the tolerance limits at a level $\gamma$ that are 100 $\beta$-percent independent of the distribution $F(x)$ and satisfy

$$Prob\left\{\left(F_{(L_2)} - F_{(L_1)}\right) \geq \beta\right\} = \gamma.$$

If we suppose $L_1 = x(R)$ and $L_2 = x(S)$, where $R < S$, the tolerant interval $[x(R), x(S)]$ is equal to the sum of elementary shares from $R$-th to $S$-th, i.e.,

$$Prob[(F_{(x_{(R)} - x_{(S)})} \geq \beta] = \gamma = \frac{N!}{(S-R-1)!(N-S+R)!} \times \int\limits_{\beta}^{1} \mathcal{Z}^{S-R-1}(1-\mathcal{Z})^{n-S-R}d\mathcal{z} =$$

$$= 1 - I_{\beta}(S - R, N - S + R + 1) = \sum_{i=1}^{S-R-1} \binom{N}{i}\beta^i(1-\beta)^{N-i}.$$

Thus, $\gamma$ is a function of arguments $N$, $S - R$, and $\beta$. There is some minimum value $N_{min}$, to which, in each specific case, a certain combination of $R$ and $S$ corresponds. It is possible to determine $\frac{1}{2}N(N-1)$ tolerant intervals with various $\gamma$ levels, among which $N/2$ and $N(N-1)/2$ (depending on whether $N$ is even or odd) will be symmetric. To ensure the symmetry of a rank, it should be connected to a condition:

$$S = N - R + 1.$$

Then, for an estimation of the cumulative distribution function $F_5(x)$ in points $x(R)$ and $x(S)$ with a confidence coefficient $\gamma$, it is possible to accept the following magnitudes:

$$F_5\left(x_{(R)}\right) = \frac{1 - \beta(R,S)}{2},$$

$$F_5\left(x_{(R)}\right) = \frac{1 + \beta(R,S)}{2}.$$

Thus, by changing value $R$ from 1 to $N/2$ and computing matching values $S$, it is possible to gain estimation $F_5(x)$ in $N$ points.

One more mode of non-parametric estimation, $F_6(x)$, can be generated from the definition of a non-parametric confidence interval $[x_{(R)}, x_{(R+K)}]$ for a quantile $x_p$ level $p$. The confidence level $\gamma$ is determined from a relation:

$$\gamma = Prob\left(\tilde{F}_6\left(x_{(R)}\right)\right) \leq p \leq \tilde{F}_6\left(x_{(R+K)}\right) = I_p(R, N - K + 1) - I_p(R + K, N - R - K + 1),$$

where $I_p(n, m)$ is Pearson's incomplete beta function:

$$I_p(n, m) = \frac{\Gamma(n+m)}{\Gamma(n) * \Gamma(m)} \int_0^P x^{n-1}(1-x)^{m-1}dx.$$

And the probability [gamma] that the quantile $x_p$ will appear between the ordinal statistics $x_{(R)}$ and $x_{(R+K)}$ does not depend on an aspect of the initial distribution $F(x)$.

The statistical relationship between the sampled value and its rank allows us to write the following approximate value [28,29]:

$$\widetilde{F}_1 = \widetilde{F}_1(x_{(R)}) = \frac{R}{N+1},\tag{22}$$

where $R$ is the rank or rank statistics (number in the ranked row) of element $x_{(R)}$.

A ranked series of ordinal statistics can also be used to estimate the correlation function of an RP.

To evaluate the correlation function in real time, the most interesting options are fairly simple rank and sign non-parametric methods of estimation [26], in particular the methods of Spearman $\rho_{sp}$ and Kendall $\rho_k$:

$$\rho_{sp}(j) = 1 - K_{sp}(N) \sum_{i=1}^{N} P_R^2 j);\tag{23}$$

$$\rho_k(j) = K_k(N) \sum_{i=1}^{N-1} R_i - 1.\tag{24}$$

Here, $P_R$ is the difference between elements $x_i$ and $x_{(i+j)}$; $K_{sp}(N)$ is Spearman's constant (at $N = const\ K_{sp} = 6/(N^3 - N)$); $R_i$ is the rank of the $i$-th element $x_i$; $K_k(N)$ is Kendall's constant (at $N$ = const, $K_k = 4/(N^2 - N)$). The procedures for estimating the correlation function according to the above formulas allow a significant simplification due to the table setting of coefficients $K_{sp}(N)$ and $K_k(N)$ and the value $P_R^2(j)$ in the microcomputer ROM (at a fixed interval of local stationarity).

## 5. Results of Computer Modeling

An analysis of the errors in estimates of the probability characteristics of a random process using the method of reflected inversions was carried out using the method of statistical modeling on a PC using Mathcad.

Consider a random function with a Gaussian distribution.

A random process, white noise $x_t$, is generated (a vector of N random numbers having a Gaussian distribution):

$$x_t = rnorm(N, \mu, \sigma).$$

A signal (trend) of the form

$$F_t = 5\left(1 - e^{(-0.01t)}\right)$$

was superimposed on a random function.

As a result, an NSRP of the form

$$y_t = x_t + F_t$$

was generated.

Figure 5 shows an example of a simulation.

In the figure, the simulated random process $y_t$ is shown in red, the trend $F_t$ is shown in blue, and the average estimate calculated by Formula (19) is shown in black.

In the figure, we have four stationary sections with a length of 9 samples, 35 samples, and 57 and 59 reports, respectively.

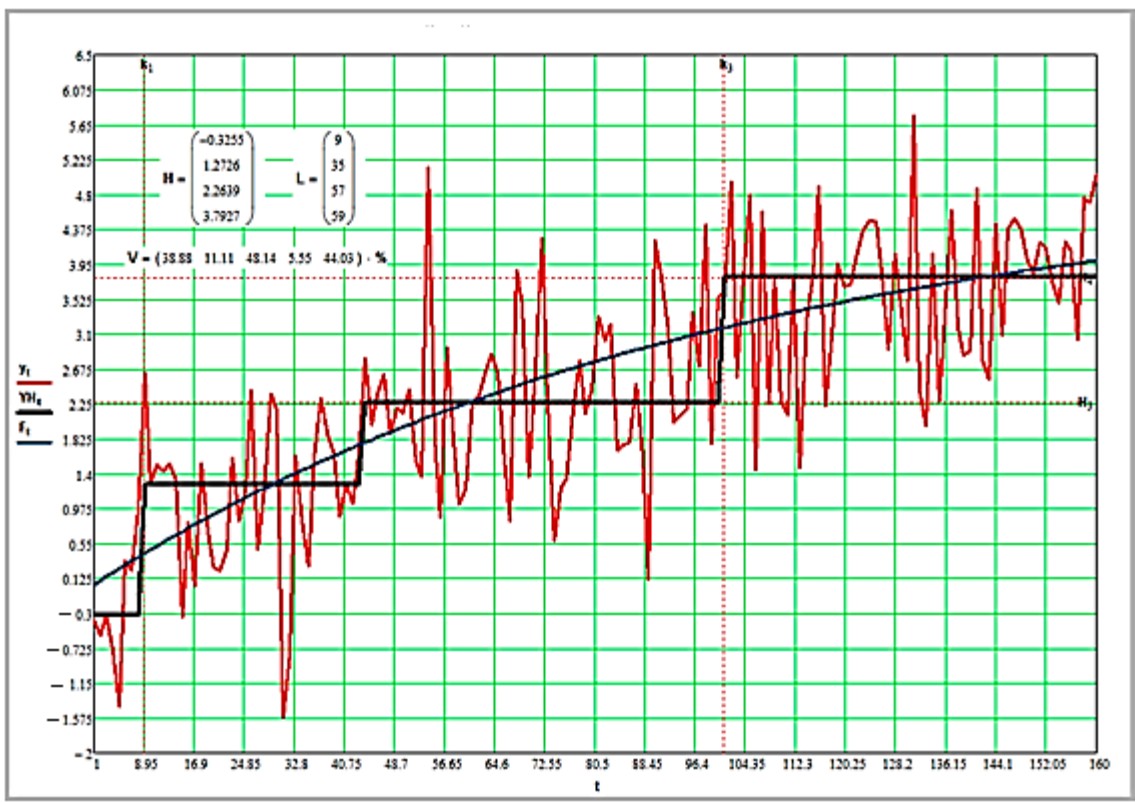

**Figure 5.** Modeling a random process with a Gaussian distribution.

The estimation of the distribution function (red; the Mathcad program designates this graph as ΦZ) and its comparison with the given one (blue; the Mathcad program designates this graph as ΦP) are shown in Figure 6.

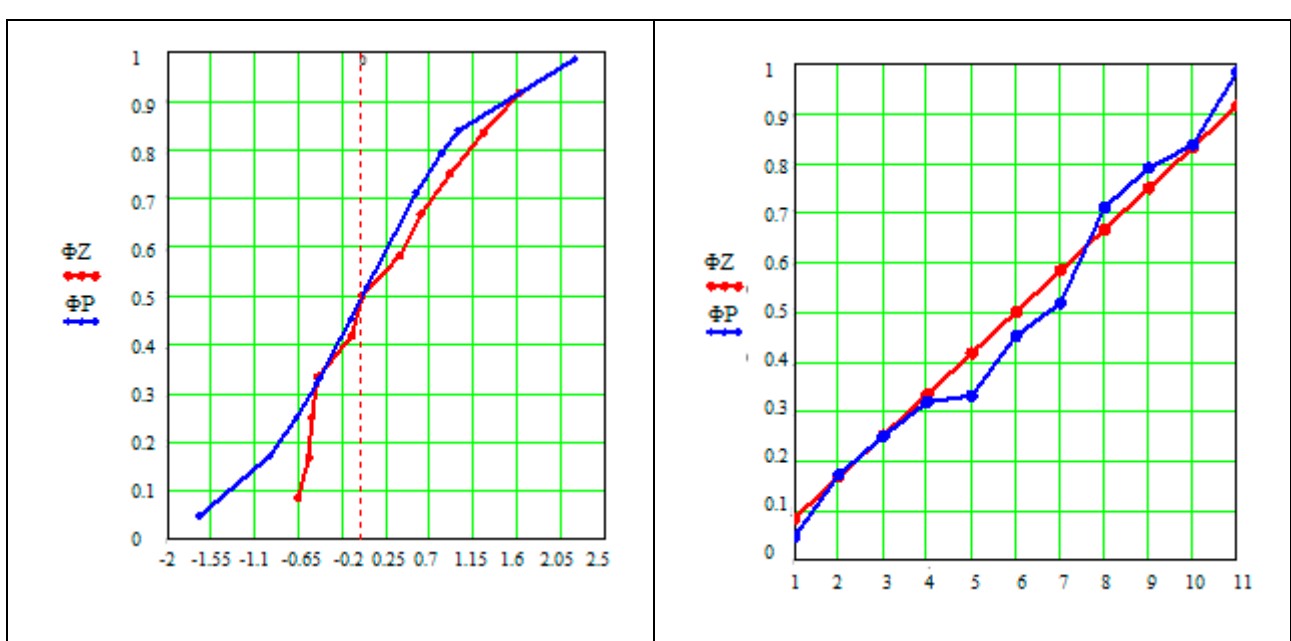

**Figure 6.** Estimation of distribution function.

Unfortunately, with this approach, it is not possible to obtain estimates of the correlation function, since obtaining the above estimates is associated with the requirement of statistical independence between the counts. Thus, an estimate of the correlation function

must be obtained separately from estimates of the other probabilistic characteristics of the RP.

To evaluate the correlation function, an RP with a correlation function of the following form was modeled:

$$R_x = \sigma^2 \cdot \exp(-\alpha|\tau|).$$

Figure 7 shows the estimate of the correlation function (red; the Mathcad program designates this graph as ΦZ) compared to the given (blue; the Mathcad program designates this graph as КΦ). The correlation function was evaluated using Formula (24).

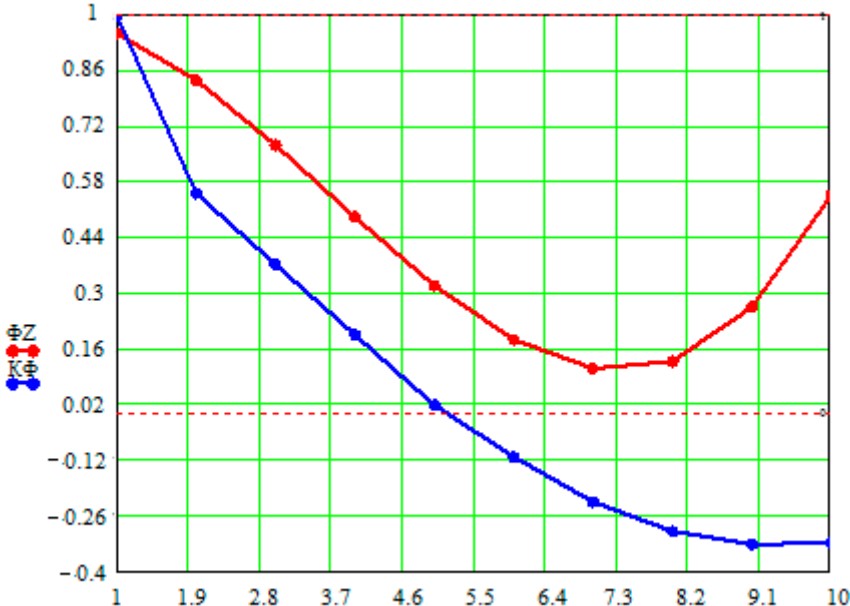

**Figure 7.** Estimation of correlation function.

The random functions with the following distribution function were also used for modeling:

rexp (*N*, *r*)—generates vector *N* of random numbers that have an exponential distribution; *r* > 0—distribution parameter (e.g., *r* = 0.9);

runif (*N*, *a*, *b*)—generates vector *N* of random numbers having a uniform distribution in which *b* and *a* are boundary points of the interval; *a* < *b* (e.g., *a* = −1, *b* = 1);

rt (*N*, *d*)—Student's distribution, where *N* is the number of random numbers, *d* is the distribution parameter, and *d* > 0.

The results of statistical modeling showed that the method is quite effective. Modeling was carried out for various types of trends: exponential, oscillatory, and linear. Also, the parameters of the algorithms varied, including the signal-to-noise ratio (the ratio of the trend amplitude to the dispersion of the random component), the sampling interval, and the value of the α significance level. The error in the estimate of the average value, as a rule, does not exceed 7%, and the variance and distribution function, 10%. Errors in correlation function estimates do not exceed 18%, which is an acceptable result for data processing purposes, for example, in the radio telemetry systems of spacecraft [30,31].

Thus, applying the best estimates of Formulas (19), (20) and (22)–(24) allows the same ranked series of ordinal statistics to be used to estimate such different probabilistic characteristics of the RP as the mean, variance, distribution function, and correlation function. This fact is very important since it allows, firstly, to significantly reduce the computational cost of obtaining these estimates, and secondly, to obtain almost complete information about the measured process in one dimension.

Of particular interest is the formation of output streams of compressed data obtained in accordance with expression (19) and their connection to the communication channel. Some aspects of this problem are covered, for example, in [32].

## 6. Conclusions

This article discusses how to evaluate the probabilistic characteristics of transient broadband random processes. Very often, a feature of RPs is that they are represented by a single implementation under conditions of a priori uncertainty about the type of distribution function. Since the use of traditional methods of mathematical statistics to calculate the probabilistic characteristics of such RPs is not possible, the use of non-parametric methods of decision theory has been proposed. The essence of the proposed methods consists in using Kendall's non-parametric statistics to divide the entire measurement interval into stationarity intervals, followed by calculating probability characteristics at each stationarity interval. By probability characteristics, we mean the mean value, variance (standard deviation), distribution function, and correlation function. To calculate probability characteristics, the ordinal and rank statistics (19)–(24) of the ranked series are used, which are very easy to calculate. It is important to keep in mind that the same ranked series is used to calculate all probability characteristics (except the correlation function). This leads to a significant reduction in computational costs since the ranking procedure is applied only once and the entire set of necessary probabilistic characteristics is calculated.

This article presents the results of computer modeling. An analysis of the errors in estimates of the probability characteristics of a random process using the method of reflected inversions was carried out using the method of statistical modeling on a PC using Mathcad. Random processes with various distribution functions, such as Gaussian distribution, exponential distribution, Student's distribution, and uniform distribution, were studied in the simulation. The function $F_t = 5\left(1 - e^{(-0.01t)}\right)$ was investigated as a trend. To evaluate the correlation function, an RP with a correlation function of the form $R_x = \sigma^2 \cdot \exp(-\alpha|\tau|)$ was modeled.

The author plans to conduct further research aimed at improving the efficiency of the developed algorithms. In particular, the algorithm for dividing the observation interval into stationarity intervals can, according to the author, be improved by using the procedure for resetting both the statistics $T^2$ and the boundary values $T^2_{min}$ and $T^2_{max}$ to zero values after determining each stationarity interval. As a result of using such a procedure, according to the author, the accuracy of determining stationarity intervals will increase. Thus, in the future, it will be necessary to conduct computer modeling with a new algorithm.

**Funding:** This research received no external funding.

**Institutional Review Board Statement:** Not applicable.

**Informed Consent Statement:** Not applicable.

**Data Availability Statement:** Not applicable.

**Acknowledgments:** The author of this article expresses sincere gratitude to the scientists and specialists of the departments "Radio-electronic systems and devices" and "Information systems and telecommunications" at N. Bauman Moscow State Technical University, whose consultations and advice were taken into account when performing scientific research, the results of which are presented in this article.

**Conflicts of Interest:** The author declares no conflict of interest.

## Abbreviations

The following abbreviations are used in this manuscript:

|       |                                      |
|-------|--------------------------------------|
| RTSSs | radio telemetry systems of spacecraft |
| NSBRP | non-stationary broadband random process |
| CSs   | computing systems                    |
| RP    | random process                       |
| SRPs  | stationary random processes          |
| NSRP  | non-stationary random process        |
| NPS   | non-parametric statistics            |
| COS   | central ordinal statistics           |
| EOS   | extreme order statistics             |

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
