# Peer review of "Determination of the Characteristics of Non-Stationary Random Processes by Non-Parametric Methods of Decision Theory"

_computation, doi:10.3390/computation11110219_

Round 1
Reviewer 1 Report
Comments and Suggestions for Authors
My impression on the paper is rather positive - the topic is interested and worth scientific research. I have, however, some critical remarks which should be taken into consideration when preparing the revised version of the manuscript:
(1) The formulas are written in different fonts, which gives the impression of carelessness.
(2) In the definition of the function sign remove the formula y/|y| (for y=0 it is invalid).
(3) Avoid abbreviations like "Let's" (line 100)
(4) In the classical statistical nomenclature the expression "null hypothesis" is used instead of "main hypothesis" (line 125)
(5) There exists Neyman-Pearson criterion, not Neumann-Pearson one (!!!) (line 130)
(5) Similarly, there exists Pearson incomplete beta function, not Prison's incomplete beta function (!!!) (line 341)
(6) Write "confidence level" with a small letter (line 341)
(7) The quality of figures is, to put it mildly, not very good.
(8) References should be formatted uniquely, as it is suitable for this journal.
(9) I strongly recommend a detailed proofreading of the whole manuscript to avoid linguistic errors and potential typos.
Author Response
Point 1: The formulas are written in different fonts, which gives the impression of carelessness.
Response 1. The remark has been corrected. All formulas are written in the same font.
Point 2: In the definition of the function sign remove the formula y/|y| (for y=0 it is invalid).
Response 2. The remark has been corrected. Formula y/|y| removed.
Point 3: Avoid abbreviations like "Let's" (line 100)
Response 3. The remark has been corrected. All the words "let's" has been deleted.
Point 4: In the classical statistical nomenclature the expression "null hypothesis" is used instead of "main hypothesis" (line 125)
Response 4. The remark has been corrected. The phrase "main hypothesis" has been corrected to "null hypothesis".
Point 5: There exists Neyman-Pearson criterion, not Neumann-Pearson one (!!!) (line 130)
Response 5. The remark has been corrected. The phrase «Neumann-Pearson» has been corrected to «Neyman-Pearson».
Point 5: Similarly, there exists Pearson incomplete beta function, not Prison's incomplete beta function (!!!) (line 341)
Response 5. The remark has been corrected. The phrase "Prison's incomplete beta function" has been corrected to "Pearson incomplete beta function"
Point 6: Write "confidence level" with a small letter (line 341)
Response 6. The remark has been corrected. The phrase "confidence level" is written in small letters
Point 7: The quality of figures is, to put it mildly, not very good.
Response 7. The remark has been corrected. Figures 1-4 are re-drawn.
Point 8: References should be formatted uniquely, as it is suitable for this journal.
Response 8. The remark has been corrected. All References are formatted according to the requirements.
Point 9: I strongly recommend a detailed proofreading of the whole manuscript to avoid linguistic errors and potential typos.
Response 9. The remark has been corrected. The text of the article has been thoroughly checked. Errors have been fixed. The introduction is written anew.
Point 1: The formulas are written in different fonts, which gives the impression of carelessness.
Response 1. The remark has been corrected. All formulas are written in the same font.
Point 2: In the definition of the function sign remove the formula y/|y| (for y=0 it is invalid).
Response 2. The remark has been corrected. Formula y/|y| removed.
Point 3: Avoid abbreviations like "Let's" (line 100)
Response 3. The remark has been corrected. All the words "let's" has been deleted.
Point 4: In the classical statistical nomenclature the expression "null hypothesis" is used instead of "main hypothesis" (line 125)
Response 4. The remark has been corrected. The phrase "main hypothesis" has been corrected to "null hypothesis".
Point 5: There exists Neyman-Pearson criterion, not Neumann-Pearson one (!!!) (line 130)
Response 5. The remark has been corrected. The phrase «Neumann-Pearson» has been corrected to «Neyman-Pearson».
Point 5: Similarly, there exists Pearson incomplete beta function, not Prison's incomplete beta function (!!!) (line 341)
Response 5. The remark has been corrected. The phrase "Prison's incomplete beta function" has been corrected to "Pearson incomplete beta function"
Point 6: Write "confidence level" with a small letter (line 341)
Response 6. The remark has been corrected. The phrase "confidence level" is written in small letters
Response 6. The remark has been corrected. The phrase "confidence level" is written in small letters
Point 7: The quality of figures is, to put it mildly, not very good.
Response 7. The remark has been corrected. Figures 1-4 are re-drawn.
Point 8: References should be formatted uniquely, as it is suitable for this journal.
Response 8. The remark has been corrected. All References are formatted according to the requirements.
Point 9: I strongly recommend a detailed proofreading of the whole manuscript to avoid linguistic errors and potential typos.
Response 9. The remark has been corrected. The text of the article has been thoroughly checked. Errors have been fixed. The introduction is written anew.

Reviewer 2 Report
Comments and Suggestions for Authors
See my report

Comments on the Quality of English LanguageSee my report
Author Response
Point 1: “ I don’t see why the paper is submitted to Computation since the computational part, essentially in Mathcad is very limited. …”
Response 1. The article is submitted to the journal "Computations" because the article is devoted to methods for computation the probabilistic characteristics of non-stationary random processes. The article considers the procedure for dividing the observation interval into stationarity intervals by computation Kendal’s statics and computation boundary values. Further, at each stationarity interval, various ways of computation the mean, variance, distribution function and correlation function are considered. Thus, the article is devoted to computation. As for modeling in the Matkad environment, the issues of estimating errors in computation probabilistic characteristics were considered here. Therefore, firstly, the submission of an article to the journal "Computations" in my opinion is justified and, secondly, there is no need to change the title of the article.
Point 2: “ Nonstationary processes are, of course, very general and cannot be reduced to the specific approach considered in this paper. There are already several definitions of stationary processes. Nonstationarity can take several forms. …”
Response 2. The article is devoted to the processing of non-stationary random processes that take place in such radio communication systems as radio telemetry systems of spacecraft. The processing of random processes in such systems is associated with a number of features. The first feature is the need for processing in real time. For example, information about processes in the combustion chamber of the spacecraft (temperature and pressure in the combustion chamber) is always a non-stationary random process. This information should be transmitted in real time so that processes in the combustion chamber can be observed on the Ground promptly. The second feature is that very often a non-stationary random process is represented by a single implementation under conditions of a priori uncertainty about the type of distribution function (for example, temperature and pressure in the combustion chamber). The third feature is that on-board computing systems always have limitations on mass-size characteristics and power consumption. In the abstract of the article, these features are indicated. Thus, the article considered methods for processing non-stationary random processes in the radio telemetry system of spacecraft with the above features. This circumstance eliminates the need to analyze a whole class of stationary and non-stationary random processes. Therefore, non-traditional processing methods are needed to process such random processes. The article considers a method for processing non-stationary broadband random processes based on the use of nonparametric methods of decision theory. These methods consist in calculating the probabilistic characteristics of non-stationary broadband random processes using nonparametric Kendall statistics. The method is based on the division of the observation interval into stationarity intervals, followed by the calculation of probabilistic characteristics for each stationarity interval. The method was developed by the author of the article and patented by the innovative patent for the invention of the Republic of Kazakhstan.
In order to make the above clear, the Introduction has been rewritten in the article.
Point 3: “In this respect, there is a whole literature on unit roots that is not even considered, …”
Response 3. Time series analysis is usually designed to recognize the type of trend. This type of analysis is possible when the entire time series is known in advance. Only in this case, the characteristic equations of the autoregressive time series model can be solved and their roots can be found. In our case, calculations are carried out in real time, i.e. at the real rate of data receipt. In other words, at each moment of time, the data measured only at this moment of time and at previous moments of time are known. The entire time series becomes known only at the end of processing of a random process. This is the specificity of the random processes under consideration. For this reason, the methods of processing random processes used in various fields, such as, for example, finance and macroeconomics, cannot be used. In general, all traditional methods of analyzing statistical characteristics of random processes are based on a priori knowledge of the type of probabilistic properties of the measured random process. For example, calculating the mean and variance by the maximum likelihood method is effective only for a Gaussian random process and is not effective for other distributions. And, if the type of distribution function is not known, then it is not possible to evaluate the effectiveness of estimating probabilistic characteristics at all. In addition, to evaluate the probabilistic characteristics of a random process by traditional methods of mathematical statistics, an implementation ensemble (at least 1000) is needed. Once again, I want to emphasize that I have developed methods for estimating (calculating) the probabilistic characteristics of non-stationary random processes represented by a single implementation, under conditions of a priori uncertainty about the type of distribution function of the measured random process. All the available methods of traditional mathematical statistics are not suitable in this case, and their analysis in the article is meaningless!
Point 4: “But most non-stationary processes do not have a correlation function and the example you take does not correspond to a non-stationary process. …”
Response 4. The reviewer's statement that "most non-stationary processes do not have a correlation function" is highly doubtful. Non-stationary random processes described by the additive-multiplicative model shown in Figure 1, as a rule, have a correlation function. For example, physical processes in the combustion chamber of a rocket engine (temperature and pressure) always have a correlation function. That is why the evaluation of the correlation function is given a place in our article. The author also did not understand the reviewer's statement that "the example you gave does not correspond to a non-stationary process." Therefore, the author finds it difficult to comment on this statement.
Point 5: “That means that for me the paper should be completely rewritten from the first (the title) to the last line (the references)”.
Response 5. Apparently, the reviewer's comments arose due to the fact that the article is not very well constructed. In this I completely agree with the reviewer. Therefore, I have rewritten the introduction, which, in my opinion, more accurately describes the meaning of the article. I also fully agree with the reviewer that some references to literature are made carelessly. This remark has been corrected.
Some details
Point 1: Line 30. There is no need to indent the paragraph after an equation since we stay in the same sentence. That mistake is made repeatedly. I will not mention it anymore.
Response 1. The remark has been corrected. The introduction has been rewritten.
Point 2: Line 32. “optimal for Gaussian distribution” should read “optimal for the Gaussian distribution”. That kind of mistake is repeated several times, for instance on lines 320, 330, 331, 340.
Response 2. The remark has been corrected. The introduction has been rewritten. Errors on lines 320, 330, 331, 340 has been corrected.
Point 3: Line 51. Mathematical variables should be in italics. That mistake is made repeatedly but randomly, see e.g., line 87. No space is needed before the opening parenthesis. The equation numbers are usually right justified. In general, the mathematical typography of this paper is horrible.
Response 3. The remark has been corrected. All formulas have been rewritten.
Point 4: Line 77. There is an erroneous horizontal bar after the curly brace. y is the argument of the mathematical function so you should write sign(y). The symbols in display equations should be of the same font and size as inline equations.
Response 4. The remark has been corrected. The formula has been rewritten.
Point 5: Line 102. “, considered” should read “, is considered”.
Response 5. The remark has been corrected.
Point 6: Line 120. I don’t understand that sentence.
Response 6. The remark has been corrected. The sentence is formulated anew.
Point 7: Line 143. “is” should read “are”.
Response 7. The remark has been corrected. "is" is replaced by "are".
Point 8: Line 184. “Kendall statistics” should read “Kendall’s statistic”.
Response 8. The remark has been corrected throughout the text.
Point 9: Line 188. “consists in calculating” should read “consists of calculating”.
Response 9. The remark has been corrected.
Point 10: Lines 193-194. This is wrong except after proper normalization, which is not mentioned at all because M[T²] and D[T²] are not detailed.
Response 10. The conclusion that the distribution of the Kendall variable at n>10 differs little from the Gaussian distribution is confirmed by literary sources. The variables M[T²] and D[T²] are not detailed, since the formulas by their definition are know-how. I can't disclose them.
Point 11: Line 200. “, since” should read “since”.
Response 11. The remark has been corrected.
Point 12: Line 201. “is indifferent how” should read “is indifferent to how”.
Response 12. The remark has been corrected.
Point 13: Lines 221 and 222. “field of stationary” should read either “field of stationary processes” or “field of stationarity”.
Response 13. The remark has been corrected. The word "stationarity" is applied
Point 14: Line 225. “an fracture point” should read “a fracture point” or “a break point”, to use a more common terminology.
Response 14. The remark has been corrected.
Point 15: "an" is replaced by "a".
Response 15. Line 230. That sentence cannot be understood.
Point 16: Line 230. That sentence cannot be understood.
Response 16. Frankly speaking, I don't understand what exactly is unclear in this sentence. If the reviewer specifies more specifically what he does not understand, I will try to answer. I ask the reviewer to indicate not only the line, but also the sentence itself, since the lines of the reviewer and I may not coincide.
Point 17: Line 265. “On Figure 4” should read “In Figure 4”.
Response 17. The remark has been corrected.

Reviewer 3 Report
Comments and Suggestions for Authors
The work is interesting in some ideas, and it can be practically useful. However, there are multiple flaws in the paper.
1. The English language should be improved. Starting from the title, what is the “Solution Theory”? The readers could only guess that it might be the decision theory.
2. The used wording is often too much tautological, for instance, the “processing random processes” is three times repeated already in the first three phrases and then again in the abstract, and throughout the text.
3. The wording “random process” is used 37 times in the paper, so it could make sense to introduce the acronym RP to facilitate the reading. Also, the stationary and non-stationary random processes can be abbreviated into SRP and NSRP.
4. The word “process” is given 69 times, which could be also diversified sometimes. The same concerns “parametric and non-parametric statistics which are repeated dozens of times.
5. The common characteristics of the mean and variance are too well-known and trivial so there is no need to present them in formulae in p. 1.
6. The grammar should be checked and corrected, for example, “vector … composed of sign statistics called a sign vector” in line 84 where should be “is called”.
7. The notations should not be used loosely. For example, in the lines 87-88 the parameter of probability is p (but it should be told explicitly; and q should be defined too) and the capital P denotes the probability (not the parameter) of the binomial event. However, in the line 89 it is written “The parameter P of this distribution”, which creates some mess.
8. Figure 1 is low informative, and Figure 2 is rather confusing than clarifying, so they can be skipped.
9. Section 2 describes some simple hypotheses testing, and it is called “Material and Methods”, although there is nothing about “material” and not clear what are the methods mentioned. Methods should be used to solve some specific problems, but the problems are described unclearly.
10. Section 3 is called “Theory/Calculations”, and it presents a lengthy description of the non-parametric Kendall’s statistics. The whole description is unnecessarily long and remains unclear what for it is considered. The non-parametric statistics are well-known in the last hundred years, so the description should be brief and answering to the question “what for is it needed”.
11. Section 4 of “Discussion and Results” actually considers some numerical simulation and its graphical presentation, but the results are not very clear.
12. All sources in the list of references are given in English, although many of them are available only in Russian. Most of those should be skipped or substituted by English analogues, because the readers of a wider audience could not all read in Russian.
Resuming, the paper is presented in a poor fashion. The work should be focused on its actual aim, and not on general descriptions. Thus, it cannot be recommended for publication in the current form. However, the authors can rewrite it in a much more precise and concise form, stating from the beginning an exact aim and describing how it was reached.
Comments on the Quality of English LanguageEnglish should be checked and corrected.
Author Response
Point 1: The English language should be improved. Starting from the title, what is the “Solution Theory”? The readers could only guess that it might be the decision theory.
Response 1. The remark has been corrected. The words "solution theory" have been replaced with "decision theory".
Point 2: The used wording is often too much tautological, for instance, the “processing random processes” is three times repeated already in the first three phrases and then again in the abstract, and throughout the text.
Response 2. The remark has been corrected. The abstract has been corrected.
Point 3: The wording “random process” is used 37 times in the paper, so it could make sense to introduce the acronym RP to facilitate the reading. Also, the stationary and non-stationary random processes can be abbreviated into SRP and NSRP.
Response 3. The remark has been corrected. The article introduces the abbreviation RP to denote a random process. The abbreviations SRP and NSRP have been introduced to denote stationary and non-stationary random processes.
Point 4: The word “process” is given 69 times, which could be also diversified sometimes. The same concerns “parametric and non-parametric statistics which are repeated dozens of times.
Response 4. The remark has been corrected. The abbreviation NPS is used to denote the phrase "non-parametric statistics".
Point 5: The common characteristics of the mean and variance are too well-known and trivial so there is no need to present them in formulae in p. 1.
Response 5. The remark has been corrected. The formula has been removed from the text.
Point 6: The grammar should be checked and corrected, for example, “vector … composed of sign statistics called a sign vector” in line 84 where should be “is called”.
Response 6. The remark has been corrected. The word "called" has been replaced with "is called".
Point 7: The notations should not be used loosely. For example, in the lines 87-88 the parameter of probability is p (but it should be told explicitly; and q should be defined too) and the capital P denotes the probability (not the parameter) of the binomial event. However, in the line 89 it is written “The parameter P of this distribution”, which creates some mess.
Response 7. The remark has been corrected. In the expression “The parameter P of this distribution”, ‘P' is written with a small letter: “The parameter p of this distribution".
Point 8: Figure 1 is low informative, and Figure 2 is rather confusing than clarifying, so they can be skipped.
Response 8. I disagree with the reviewer. Figure 1 clearly depicts an additive-multiplicative signal model. Figure 2 explains the verification of the hypothesis about the symmetry of the distribution of a random function. I think that Figure 1 and Figure 2 should remain.
Point 9: Section 2 describes some simple hypotheses testing, and it is called “Material and Methods”, although there is nothing about “material” and not clear what are the methods mentioned. Methods should be used to solve some specific problems, but the problems are described unclearly.
Response 9. The name of the section “Material and Methods” is set in the template, i.e. this name is a requirement for the authors. We can't name the sections ourselves.
Point 10: Section 3 is called “Theory/Calculations”, and it presents a lengthy description of the non-parametric Kendall’s statistics. The whole description is unnecessarily long and remains unclear what for it is considered. The non-parametric statistics are well-known in the last hundred years, so the description should be brief and answering to the question “what for is it needed”.
Response 10. The "Theory/Calculations" section does not provide a "lengthy description of the non-parametric Kendall's nonparametric statistics". This section presents the algorithm I developed for dividing the observation interval into stationarity intervals using non-parametric Kendall’s statistics. The developed algorithm is the basis for calculating the probabilistic characteristics of non-stationary random processes represented by a single implementation in conditions of a priori uncertainty about the form of the distribution function. The algorithm is patented by an innovative patent for an invention in the Republic of Kazakhstan.
Point 11: Section 4 of “Discussion and Results” actually considers some numerical simulation and its graphical presentation, but the results are not very clear.
Response 11. The article is devoted to the development of an algorithm for calculating the probabilistic characteristics of non-stationary random processes. It is shown how, with the help of non-parametric Kendall’s statistics, it is possible to divide the time series of observations into stationarity intervals. The algorithms for calculating probabilistic characteristics at each stationarity interval are given below. But, it is not enough to present only the calculation algorithms. It is important to study the errors of the developed algorithms, since the errors give an idea of the effectiveness of the developed algorithms. The study of errors is the final part of the article. Therefore, these errors are shown in the last section “Discussion and results". Errors in the calculation of probabilistic characteristics are demonstrated in the form of graphs that are the result of modeling in a Mathcad environment. Numerical values of errors in estimating the mean value, variance, distribution function and correlation function are also given.
Point 12: All sources in the list of references are given in English, although many of them are available only in Russian. Most of those should be skipped or substituted by English analogues, because the readers of a wider audience could not all read in Russian. Resuming, the paper is presented in a poor fashion. The work should be focused on its actual aim, and not on general descriptions. Thus, it cannot be recommended for publication in the current form. However, the authors can rewrite it in a much more precise and concise form, stating from the beginning an exact aim and describing how it was reached. Comments on the Quality of English Language English should be checked and corrected.
Response 12. I am a Russian-speaking author. Therefore, when conducting research, I studied mainly Russian-language literature. In the article, I am obliged to make references to those literary sources that I have studied.
Conclusion. In order to make the purpose of the article more understandable, I have rewritten the "Introduction", where the purpose of the article is clearly indicated. I hope that the new edition of the Introduction makes it clear that the purpose of the work is to develop methods for processing non-stationary broadband random processes represented by a single implementation under conditions of a priori uncertainty about the form of the distribution function.

Reviewer 4 Report
Comments and Suggestions for Authors
In 'Determination of the characteristics of non-stationary random processes by non-parametric methods of decision theory', the author discusses a method of processing non-stationary broadband random processes based on the use of non-parametric methods of decision theory and proposes an algorithm to divide the observation interval into stationary intervals. According to the revision, I reckon this manuscript is informative to the audience of Computation. I have some following minor concerns before the potential publication, hoping it helps.
1. Some figures are still not clear enough. For example, (i) I would suggest a more clear illustration of the sub-figures in Figure 1. What each subplot means should be demonstrated in the main text. (ii) In the caption of each figure, I would suggest a further introduction about the figure, e.g., the parameters, the meaning, or the crucial conclusion. (iii) Please improve the dpi of Figures 1, 2, 3, 5, 7 because there is obvious serrated shape. Figure 4 is fine.
2. As the author mentioned, he considered an algorithm to divide the observation interval into stationary intervals. I would suggest the author highlight this in the main manuscript in the algorithm form and format.
3. Please clarify the origin of Eqs. 16 and 17 more clearly with citation or demonstration.
4. The formation of equations in this manuscript should be thoroughly checked and unified, e.g., in line 295, page 5, the term Z>C_1 has a different formation compared to lines 296 and 297.
5. In Conclusions (Sec. 5), the author should also mention the potential improvement of the current method, e.g., the disadvantages and future directions.
6. In Eq. 4, the author may make a clerical error. The combination operator C_i^n in line 235 may be revised as C_n^i since the Bernoulli event happens i times in n independent tests.
7. The author may mistakenly write 0.5 as 0,5 in lines 290 and 292, as well as in Eq. 11. Please check it out carefully.
Author Response
Remark 1. Some figures are still not clear enough. For example, (i) I would suggest a more clear illustration of the sub-figures in Figure 1. What each subplot means should be demonstrated in the main text. (ii) In the caption of each figure, I would suggest a further introduction about the figure, e.g., the parameters, the meaning, or the crucial conclusion. (iii) Please improve the dpi of Figures 1, 2, 3, 5, 7 because there is obvious serrated shape. Figure 4 is fine.
Response. The remark has been corrected. The text after Figures 1, 2, 3, 4 provides an explanation for each figure. For Figures 5-7, such explanations already exist in the text. Figures 1-3 are redrawn. It is not possible to redraw Figures 5, 7, since they are made by the Mathcad program. Nevertheless, the author has tried to improve their quality.
Remark 2. As the author mentioned, he considered an algorithm to divide the observation interval into stationary intervals. I would suggest the author highlight this in the main manuscript in the algorithm form and format.
Response. The remark has been corrected. The algorithm for dividing the observation interval into stationarity intervals in the article is highlighted in a separate chapter "3. Division of the observation interval into stationarity intervals using Kendall's statistics". New chapter titles were also used:
"2. Nonparametric statistics and their use for processing random processes"; "4. Evaluation of probabilistic characteristics of a random process using ordinary and rank statistics"; "5. Results of computer modeling".
Remark 3. Please clarify the origin of Eqs. 16 and 17 more clearly with citation or demonstration.
Response. The remark has been corrected. For formulas 16 and 17 , the following explanation is given in the article: «To calculate and , it is necessary to know the average value and variance of Kendall’s statistics. The formulas for calculating the mean and variance of Kendall’s statistics are known to the author, but they are know-how and therefore are not given in the article. The value of the percentage point is taken for the Gaussian distribution, since the binomial distribution is well approximated by the Gaussian distribution».
Remark 4. The formation of equations in this manuscript should be thoroughly checked and unified, e.g., in line 295, page 5, the term Z>C_1 has a different formation compared to lines 296 and 297.
Response. The remark has been corrected. The designation is the same everywhere.
Remark 5. In Conclusions (Sec. 5), the author should also mention the potential improvement of the current method, e.g., the disadvantages and future directions.
Response. The remark has been corrected. In the conclusions, information has been added about the potential improvement of the current method of dividing the observation interval into stationarity intervals.
Remark 6. In Eq. 4, the author may make a clerical error. The combination operator C_i^n in line 235 may be revised as C_n^i since the Bernoulli event happens i times in n independent tests.
Response. The remark has been corrected. The formula has been corrected.
Remark 7. The author may mistakenly write 0.5 as 0,5 in lines 290 and 292, as well as in Eq. 11. Please check it out carefully.
Response. The remark has been corrected. All numbers 0,5 are forwarded to 0.5.

Round 2
Reviewer 2 Report
Comments and Suggestions for Authors
See my report

Comments on the Quality of English LanguageSee my report
Author Response
Remark. «I still don’t see why the paper is submitted to Computation since the computational part, essentially in Mathcad is very limited».
Response. It is unclear why the reviewer makes this remark again. I have already answered this question. The reviewer just had to study my answer carefully. I have to respond to this remark again. The article is submitted to the journal "Computations" because the article is devoted to methods for computation the probabilistic characteristics of non-stationary random processes. The article considers the procedure for dividing the observation interval into stationarity intervals by computation Kendal’s statics and computation boundary values. Further, at each stationarity interval, various ways of computation the mean, variance, distribution function and correlation function are considered. Thus, the article is devoted to computation. As for modeling in the Matkad environment, the issues of estimating errors in computation probabilistic characteristics were considered here. Therefore, firstly, the submission of an article to the journal "Computations" in my opinion is justified and, secondly, there is no need to change the title of the article. Thus, the article is submitted to the Journal of Computation, because the article is devoted to computing.
Remark. «….there is no coherency with the in-line symbols (they are in italics in the equations whereas in Roman in the text; they should be in italics) ».
Response. Since the designation of characters in the text in italics improves the perception of the entire text, we can agree with the reviewer. All characters in the text will be indicated in italics.
Remark. «… the title (much too broad), the abstract where the focus is more limited to “broadband” processes (that I don’t know), and finally the paper which is much more specific do not fit together.
Response. The problem of the reviewer, apparently, is that the reviewer did not carefully read the article and did not understand its essence. With careful reading of the article, any researcher will understand the logical connection between the title, annotation, introduction and the main part of the article. The title of the article indicates which problem is being solved in it. It solves the problem of computing the probabilistic characteristics of random processes. The abstract and the introduction explain absolutely unambiguously why and where the problem arises with computing the probabilistic characteristics of random processes. The introduction also explains why nonparametric methods of decision theory should be used. The main part reveals how this problem can be solved. Maybe the reviewer believes that the title of the article should be, for example, as follows: "Determination of the characteristics of non-stationary random processes represented by a single implementation under conditions of a priori uncertainty about the type of distribution function, by nonparametric methods of the theory of decisions for signal processing in radio telemetry systems of spacecraft." Such a title, of course, better explains the essence of the article. But, it's too bulky. It is not customary to use such names in articles. The name should be concise. The title should reflect the main purpose of the article. An abstract and an introduction are needed to explain what the essence of the article is. In my opinion, the title, abstract, introduction and the main part of the article in a logical sequence reveal the essence of the article.
Remark. «This time, because of the track changes in the text, the paper is nearly unreadable and I cannot check the grammar».
Response. All grammatical errors pointed out by reviewers are corrected. If the reviewer cannot check the grammar, then I do not know how to react to it.
Remark 1. In general, a nonstationary random process does not have a constant mean (except if it is zero) and a constant variance, and, consequently, the correlation function is undefined. Let us first suppose a zero mean to simplify. The correlation is based on the covariance (??)??(??−??)? and the time-dependent variance but it does not depend on k alone but also t. See e.g., Priestley (1988). Otherwise, the process is stationary in the second order. A priori, contrary to a strictly stationary process, every (??) has its own distribution so I don’t understand what you are trying to estimate.
Response. The reviewer is always trying to understand the essence of the article from the point of view of the methods of classical mathematical statistics. In the abstract and in the introduction it is clearly written that the methods of classical mathematical statistics are not suitable for calculating the probabilistic characteristics of non-stationary random processes in signal processing in radio telemetry systems of spacecraft. The peculiarity of such signals is that they are represented by a single implementation under conditions of a priori uncertainty about the form of the distribution function. Therefore, it is meaningless to assert that "each (??) has its own distribution". This would be correct if we had a large number of random process implementations. I recommend the reviewer to read the introduction carefully again. As for the statement "I don't understand what you are trying to evaluate," I explain again. The article considers a method for estimating the probabilistic characteristics (mean, variance, distribution function, correlation function) of non-stationary random processes represented by a single implementation under conditions of a priori uncertainty about the type of distribution function. To estimate the probabilistic characteristics of nonstationary random processes, nonparametric methods of decision theory are used in the article. No other methods can be applied to the processing of broadband non-stationary random processes in the radio telemetry systems of spacecraft!
Remark 2. If the mean is not zero, you should work with deviations with respect to the mean but the method you use (breaking the series in so-called stationary intervals) is surely not good. There exist nonparametric regression methods for that purpose, using a kernel estimator, that you seem to ignore, for instance, Vogt (2012), and some can be used recursively.
Response. Absolutely incorrect statement of the reviewer! This statement confirms my conclusion that the reviewer inattentively read the article. It is the method I use that is the only possible one for estimating the probabilistic characteristics of broadband non-stationary random processes represented by a single implementation under conditions of a priori uncertainty about the form of the distribution function. There are no other methods for estimating broadband non-stationary random processes in the radio telemetry systems of spacecraft. All methods based on regression analysis require that the entire time series be known in advance. In our case, this is not the case. I gave the corresponding explanation to the reviewer, responding to his Remarks last time. I recommend that the reviewer carefully read the author's answers to his comments.
Remark 3. Nearly everywhere, there is confusion between the process and its realization, or a population characteristic and the estimate. There is no such thing as the measurement of a stochastic process or a measured random process.
Response. Once again, I am convinced that the reviewer inattentively read the article. There is no confusion in terminology. The article repeatedly explains that we consider non-stationary random processes represented by a single implementation under conditions of a priori uncertainty about the form of the distribution function. The reviewer is very wrong in claiming that " There is no such thing as .... a measured random process." A non-stationary broadband random process transmitted to Earth by the telemetry system of spacecraft is a measured random process.
Remark 4. Contrary to what you say in several places, most work (if not all) on time series is done on a single realization of the process. Anyway, your references do not cover the huge existing literature on time series. None of your references has “time series” in the title.
Response. I already answered this question last time. Apparently, the reviewer didn't read my answer. I have to explain again. To analyze time series, it is necessary that the entire time series is known in advance. In our case, calculations are carried out in real time, i.e. at the real rate of data receipt. In other words, at each moment of time, the data measured only at this moment of time and at previous moments of time are known. The entire time series becomes known only at the end of processing of a random process. This is the specificity of the random processes under consideration. It makes absolutely no sense to analyze time series in our case! For this reason, we have not studied the literature on time series.
Remark 5. What is true is that many papers present Monte Carlo results with many replications of the data-generating process, not like you did the simulation of a single series that has no empirical value. I wonder if your computational tool, MathCad, can handle thousands of time series.
Response. The article is aimed at calculating the probabilistic characteristics of non-stationary random processes represented by a single implementation under conditions of a priori uncertainty about the form of the distribution function. Considering "thousands of time series" does not make any sense to us.
Remark 6. What seems to be more specific in your case is that you want to use on-line methods, also called recursive methods, for instance, Ljung & Söderström (1981) or Young (2011). Your references on that subject are also lacking.
Response. I won't repeat myself. See the answer to the remark above.
Remark 7. Your assessment that “For example, well-known maximum likelihood methods are effective only for a Gaussian RP and are not effective for other distributions. They turn out to be completely ineffective if the type of the distribution function is not known a priori” is wrong. It is known that, in most cases, the estimators of the parameters are still consistent and asymptotically normal although they are not asymptotically efficient; i.e., of minimum variance. It is even true for time-dependent models. I have seen no indication that noise is not normal. At least not in your simulations of Section 4.
Response. My statement is that "For example, the well-known maximum likelihood methods are effective only for Gaussian RP and are not effective for other distributions. They turn out to be completely ineffective if the type of distribution function is not known a priori " is well known in the Russian-speaking scientific community and is an axiom that does not need to be proved to anyone. I recommend the reviewer to familiarize himself with the following works:
- David. Ordinal statistics. Science: Moscow, Russia, 1979; 336 p. [in Russia]
- Ya. Vilenkin. Statistical processing of the results of the study of random functions. Moscow: Energy; 1979. [in Russia]
Further, it is unclear what the noise has to do with it. If we are talking about white noise, then, of course, it has a normal distribution. But, we are considering real random processes. We don't care what their distribution is. The distribution of measured random processes may be a priori unknown.
Remark 8. There is no theoretical proof for your method and, as said above, a single simulation is not enough.
Response. The theoretical proof of our method is presented in a huge number of works devoted to nonparametric methods of decision theory. I bring to the attention of the reviewer that I have developed a method for dividing the observation interval into stationarity intervals, which is patented by an innovative patent for the invention of the Republic of Kazakhstan. The validity of this method is confirmed by the results of mathematical modeling. Another method of confirmation can only be stand tests or the introduction of our algorithms into real telemetry systems of spacecraft. For obvious reasons, we cannot organize either the first or the second. There are no other ways to confirm. If the reviewer is aware of any other methods of proving the validity of the methods developed by us, please report them. I will definitely use them in my research.
Remark 9. I see no relation between your method and radio-communication systems, telemetry, etc. What do you do with these probabilistic characteristics that you estimate (badly, in my opinion)?
Response. The reviewer should carefully read the article, especially the introduction. The method developed by us was developed specifically for use in telemetry systems of spacecraft. Telemetry systems of spacecraft belong to the class of radio-communication systems.
Remark 10. You seem to like procedures based on ranks. They are widely used in time series analysis, for instance, for testing autocorrelation, but again there is no reference in that area. See e.g. Hallin & Puri (1992).
Response. The point is not that I like ranks, but that the use of nonparametric methods of decision theory is the only possible method for estimating the probabilistic characteristics of broadband nonstationary random processes, represented by a single implementation under conditions of a priori uncertainty about the form of the distribution function. I have already given my explanations about the analysis of time series. It doesn't make sense to repeat yourself.
Remark. Please avoid all your confusing abbreviations.
Response. The use of abbreviations was strongly recommended by Reviewer 3. Here is one of his remarks: "The wording "random process" is used 37 times in the paper, so it could make sense to introduce the acronym RP to facilitate the reading. Also, the stationary and non-stationary random processes can be abbreviated into SRP and NSRP». Reviewer think abbreviations are confusing. Perhaps he's right right. But, I cannot fulfill the mutually exclusive requirements of two reviewers. Let the editor of the journal make a decision. As he advises, so I will do.
Remark. Despite you changing most of the bibliography, your citations are still not appropriate, e.g. [9] that I recommended to you has no relation to what you mentioned: “If we take into account that the number of measured sources of information can be several tens or even hundreds”. I must confess I am not familiar with most of your references.
Response. The conclusion that "... the number of measured sources of information can be several tens or even hundreds" follows from the general concept of the article [9]. What other citations does the reviewer consider inappropriate? Give me an example and I'll think about how to fix it.
Remark. To conclude. I cannot recommend your paper with a wrong title, a wrong abstract, and a procedure that lacks justification. Usually, the reasons to use nonparametric statistics are to avoid distributional assumptions (but you work with normal deviates) and avoid the effects of outliers (but in Eq. (20) you use the extreme values to estimate the variance! the worst estimator). The paper has nothing to offer in terms of computation, the focus is too general, the literature is badly covered and the given references are useless, there is no comparison with the multiple methods in the literature on multiple trend breaks and locally stationary processes.
Response. When the reviewer says that the title of the article, the abstract and the procedures are incorrect, it only indicates that the reviewer is trying to impose his vision on the author about how the article should be constructed. This is a manifestation of disrespect for the author of the article. Instead, the reviewer should carefully read all sections of the article, as well as my responses to his initial Remarks. Nonparametric statistics are used in processing random processes whose distribution function is not known. Nonparametric statistics have a known distribution. This allows them to be used to test various statistical hypotheses, for example, the hypothesis of stationarity. I don't understand the statement "(but you work with normal deviates)". If the reviewer explains, I will give my Remark. It is not clear what the reviewer's statement is based on, that extreme values for estimating variance are the worst estimator. What research has the reviewer done to make such claims? The results of mathematical modeling carried out by us proved that it is the extreme values for estimating the variance that are the best estimator. Maybe the reviewer is referring to the data received on the receiving side after transmitting them over the communication channel? In this case, indeed, the communication channel may distort the data. Then the extreme values may be faulty. There are methods for determining whether extreme values are faulty or not. If the extreme values are faulty, they are either deleted or corrected. We are dealing with data coming from the sensors of the telemetry system. There can be no faulty data in these signals, unless, of course, the sensor is serviceable. The reviewer's statement that " The paper has nothing to offer in terms of computation" is erroneous. The whole article is devoted to computations. It is very strange that the reviewer does not see this. The statement that "the literature is poorly covered, and the references given are useless" is obviously based on the fact that the reviewer once again wants the article to have a time series analysis. But, I will not Remark on this anymore, since I have already spoken a lot about this.
Some details
Remark. Lines 41-42. Please define a broadband process. If you mean “rapidly changing”, the examples you took in Figure 1 or 5 are not rapidly changing.
Response. Erroneous statement of the reviewer. Fig. 1 and Fig. 5 show exactly fast-changing (rapidly) random processes.
Remark. Line 148. See above.
Response. On line 148, I have the following sentence: «We introduce the null hypothesis about the symmetry of the distribution». It is not clear what I should Remark on here. I ask the reviewer not to indicate the line number, but to quote a sentence that causes the reviewer to misunderstand. This is due to the fact that the lines of me and the reviewer, apparently, do not match.
Remark. Line 163. See above.
Response. On line 163, I have the following sentence: ««Such a decisive rule is unbiased only for a P > 0.5. At P < 0.5, the decisive rule Z < C2 turns out to be…». It is not clear what I should Remark on here. I ask the reviewer not to indicate the line number, but to quote a sentence that causes the reviewer to misunderstand. This is due to the fact that the lines of me and the reviewer, apparently, do not match.
Remark. Figure 1. The top plot is wrong (there are points with adjacent segments having both a negative slope) and cannot correspond to the plot at the bottom.
Response. The point of Figure 1 is to show an additive-multiplicative signal model. In such a model, the measured signal consists of a non-stationary trend F(t) and a stationary component x(t). Figure 1 demonstrates this model well. The fact that the top plot cannot correspond to the plot at the bottom is a disadvantage of drawing graphs. But, this is not an obstacle to understanding the additive-multiplicative signal model. However, I have redrawn Figure 1.
Remark. Line 218. Mathematical variables should be in italics. That mistake is made repeatedly but randomly, see e.g., line 231. No space is needed before the opening parenthesis. Already noticed this on the submission. Mathematical functions like sign and Prob are usually typed in Roman.
Response. All formulas are written in italics. Mathematical symbols in the text will also be executed in italics. Spaces are eliminated everywhere before the opening parenthesis. The mathematical functions sign and Prob are written using a formula editor that sets the font itself.
Remark. Lines 386-387. M[T²] and D[T²] are undefined. Already noticed this on the submission.
Response. I have already given the reviewer the following answer to this remark: “The variables M[T2] and D[T2] are not detailed, since the formulas by their definition are know-how. I can't disclose them”.
Apparently, the reviewer didn't read my answer. I have nothing to add.
Remark. Line 799. This reference is surely wrong. Routledge is the commercial editor, not the author. The book is out of print, so even if I had time for this review, I could not order it.
Response. The remark has been corrected. The author is Bob Marley.
Remark. Line 800. Same remark. The author (obtained by clicking on the link) is Peter Zörnig.
Response. The remark has been corrected.

Reviewer 3 Report
Comments and Suggestions for Authors
The paper has been improved in revision, but there are some issues to improve.
1. In Template, the Sections names are given as examples, so they can be changed due to the meaning of what they actually present. The Section 2 describes some simple hypotheses testing, so to keep it called “Material and Methods” makes no sense. Similarly, other sections should be named meaningfully.
2. As I noted previously, “all sources in the list of references are given in English, although many of them are available only in Russian. Most of those should be skipped or substituted by English analogues, because the readers of a wider audience could not all read in Russian.” However, the author answers that he studied the works in Russian language, so he can give the references on them. In this case, I suggest to mark the sources when needed as “(in Russian)”, then the non-Russian-language readers would not look for those sources.
Comments on the Quality of English Languageshould be checked again.
Author Response
Remark 1. In Template, the Sections names are given as examples, so they can be changed due to the meaning of what they actually present. The Section 2 describes some simple hypotheses testing, so to keep it called “Material and Methods” makes no sense. Similarly, other sections should be named meaningfully.
Response. The remark has been corrected. The algorithm for dividing the observation interval into stationarity intervals in the article is highlighted in a separate chapter "3. Division of the observation interval into stationarity intervals using Kendall's statistics". New chapter titles were also used:
"2. Nonparametric statistics and their use for processing random processes"; "4. Evaluation of probabilistic characteristics of a random process using ordinary and rank statistics"; "5. Results of computer modeling".
Remark 2. As I noted previously, “all sources in the list of references are given in English, although many of them are available only in Russian. Most of those should be skipped or substituted by English analogues, because the readers of a wider audience could not all read in Russian.” However, the author answers that he studied the works in Russian language, so he can give the references on them. In this case, I suggest to mark the sources when needed as “(in Russian)”, then the non-Russian-language readers would not look for those sources.
Response. 2. The remark has been corrected. All sources in Russian are marked as [in Russia].
